# Effects of Agronomic Practices on the Severity of Sweet Basil Downy Mildew (*Peronospora belbahrii*)

**DOI:** 10.3390/plants10050907

**Published:** 2021-04-30

**Authors:** Chen Omer, Ziv Nisan, Dalia Rav-David, Yigal Elad

**Affiliations:** 1Department of Plant Pathology and Weed Research, Agricultural Research Organization, The Volcani Center, Bet Dagan 50250, Israel; chen.chermon@gmail.com (C.O.); zivnisan@gmail.com (Z.N.); dalia@volcani.agri.gov.il (D.R.-D.); 2Agroecology and Plant Health, Robert H. Smith Faculty of Agriculture, Food and Environment, The Hebrew University of Jerusalem, Rehovot 7610001, Israel

**Keywords:** agrotechnical control, cultural control, integrated management, downy mildew, greenhouse crop, plant disease, *Ocimum basilicum*, *Peronospora belbahrii*

## Abstract

Downy mildew (caused by *Peronospora belbahrii*) is a severe disease of sweet basil (*Ocimum basilicum*) crops around the world. We examined cultural methods for reducing the severity of sweet basil downy mildew (SBDM) under commercial conditions in greenhouses and walk-in tunnels. The effects of the orientation of walk-in tunnels, air circulation in greenhouses, plant density, and soil mulch were tested. SBDM was less severe in the tunnels that were oriented north-south than in those oriented east-west, but the yields in both types of tunnels were similar. Increased air circulation reduced SBDM severity, but did not affect yield. Gray or transparent polyethylene mulch reduced SBDM severity and, in most cases, increased yield relative to bare soil/growth medium. Yellow polyethylene mulch provided a smaller amount of control. The combination of increased air circulation and yellow polyethylene mulch provided synergistic SBDM control, whereas no synergism was observed when we combined increased air circulation with the other two types of mulch. Planting at half the usual density reduced disease severity. The reduced plant density was associated with reduced yield in the greenhouses, but not in the tunnels. All of the tested methods provided an intermediate level of SBDM control that varied among the different experiments.

## 1. Introduction

Cultural methods (i.e., agricultural practices) can play important roles in preventing or minimizing many plant diseases. Different practices can be implemented during seeding/planting, as the crop is growing or even after harvest. Cultural techniques aimed at minimizing disease include the choice of altitude at which to plant a particular crop, the means of preparing and cultivating the soil, the use of particular cultivars, the treatment of propagation material, the choice of particular planting times and depths, the exposure to air movement, the direction of plant rows, weed-management and irrigation practices, changes in plant nutrition, and general sanitation [1,2,3]. In covered crops, farmers can also consider the type of cover spread over the crop, heating, ventilation, and soil mulch [4]. Cultural methods that affect the environmental conditions inside the greenhouse and minimize the presence of water on the canopy can be effective ways of controlling diseases that are promoted by high humidity [5].

Sweet basil (*Ocimum basilicum* L.) is an economically important annual herb crop from the Labiatae family that is grown in polyethylene-covered structures (i.e., greenhouse structures or walk-in tunnels). Sweet basil greenhouses are common along the ridge above the Syrian-African Rift, south of the Sea of Galilee, and around and north of the Dead Sea. Crops are planted from September on, so winter and spring crops are common, but are challenged by humidity-promoted diseases [6,7,8,9].

Downy mildew of basil (*Peronospora* sp.) was first identified in Uganda [10]. The causal agent was identified as an oomycete pathogen and named *Peronospora belbahrii* Thines [11,12]. This pathogen mainly infects the leaf blades of sweet basil [12]. Infected leaves are distorted, asymmetric chlorosis develops on the leaf blades, and dark spores (on sporangiophores) form on the underside of the leaves [13,14]. Sweet basil downy mildew (SBDM) was found to be particularly severe when plants were kept wet for at least 6 to 12 h immediately after inoculation; the pathogen was most active at 20 °C, whereas at 12 and 27 °C, the disease was suppressed [13]. Sporulation occurs when infected plants are incubated for at least 7.5 h in a dark, moisture-saturated atmosphere at 10 to 27 °C and is suppressed by light [14]. The effects of temperature and relative humidity on SBDM have been confirmed under field conditions. Passive heating has been shown to increase root temperatures and induce resistance in sweet basil [9].

Methods for controlling SBDM may involve fungicides, seed treatments, and/or breeding for resistance [15,16,17]. Seeds have been found to be vehicles for pathogen transmission. It was, therefore, recommended that a seed-certification scheme be established and that the pathogen be controlled on the seeds [18]. In terms of fungicide treatments, mefenoxam with copper hydroxide, azoxystrobin, and mandipropamid effectively suppress SBDM, but the economic value of those treatments has not been demonstrated [19]. The development of resistance [15] to chemical fungicides and the need to avoid any residues on the harvested shoots limit the use of fungicides in sweet basil.

Previous studies have examined other humidity-promoted diseases of sweet basil: *Botrytis cinerea*-induced gray mold and *Sclerotinia sclerotiorum*-induced white mold. Gray mold initially infects stem wounds after harvest and white mold initially infects the stem base. Both of those pathogens are necrotrophs that benefit from the film of water present on the plant organs. Increased plant spacing, the use of polyethylene mulch, and increased greenhouse aeration have been shown to reduce the incidence of both gray mold and white mold in sweet basil [7,8]. *P. belbahrii* is an oomycete biotroph that initially infects leaves. This makes it different from the two necrotrophic pathogens, but it is similar to those pathogens in that it depends on water present on the leaf for infection. Our hypothesis was that the conditions in sweet basil greenhouses are favorable for SBDM infection and that we could affect the severity of disease by manipulating those conditions. Thus, the aim of the present research was to test cultural means of controlling SBDM in greenhouses and walk-in tunnels under commercial-like conditions. The cultural means examined included increased air circulation, manipulation of the direction of the walk-in tunnels, decreased planting density, and the use of polyethylene mulch.

## 2. Results

### 2.1. Tunnel Orientation

The effect of tunnel orientation on SBDM could only be evaluated at Site 2. We found that SBDM was less severe in the walk-in tunnels that were oriented north-south, as opposed to east-west. The decrease in severity varied with the experiment, soil cover, and year (Appendix A). In terms of the area under disease progress curve (AUDPC), the difference in SBDM severity between east-west and north-south tunnels was between 25 and 33% in the first year and was about 63% in the second year (Table 1 and Appendix A). The shoot yields in the two types of walk-in tunnels were significantly similar to each other (Table 1). Thus, the north-south tunnel direction is associated with lower SBDM severity as compared with east-west-oriented tunnels.

### 2.2. Effect of Increased Air Circulation

The effect of increased air circulation (AC) was evaluated only in the greenhouses at Site 1. We increased the air circulation in those greenhouses by operating fans during the night. This increased air circulation reduced SBDM severity (Table 2 and Appendix A). Disease reduction reached up to 72.5% in all experiments conducted over three seasons (Table 2 and Appendix A), but did not affect shoot yield (Table 2). Therefore, AC decreases SBDM. Additionally, the results for the combination of increased air circulation with polyethylene mulch are described below.

### 2.3. Polyethylene Mulch

In greenhouses at Site 1, covering the beds with gray or transparent polyethylene decreased SBDM severity (Table 3 and Appendix A). Disease severity was significantly reduced (by up to 65%) in six out of the seven evaluation comparisons (Appendix A). A whole-season evaluation based on AUDPC revealed 13‒64% disease reduction in the covered beds (Table 3 and Appendix A). The use of polyethylene mulch significantly increased yield in five of the six greenhouse experiments (Table 3 and Appendix A). Therefore, polyethylene mulch can reduce SBDM and increase yield.

Three types of polyethylene mulch were compared in two growing seasons. The gray and the transparent polyethylene mulch provided more consistent and effective disease control (Table 4). The combination of increased air circulation and polyethylene mulch was examined in a third experiment (Figure 1, Table 5), which revealed no interaction between those treatments. Disease was significantly reduced by the increased air circulation and the gray polyethylene. The results observed for the transparent polyethylene were significantly different from those observed for the yellow polyethylene (Figure 1, Table 5). Calculating the synergy factor (SF) of the reduction in severity provided by each of the polyethylene mulches and the air-circulation treatment revealed no synergism between increased air circulation and either the gray or the transparent polyethylene. However, there was a significant synergistic effect between the increased air circulation and the yellow mulch (SBDM severity: SF = 1.04; AUDPC: SF = 1.12).

The effect of polyethylene mulch was also evaluated at Site 2. SBDM severity was significantly reduced in six out of eight comparisons (20‒52% disease reduction) and in seven out of the eight epidemic (AUDPC) comparisons (17‒51% reduction; Table 6 and Appendix A). Shoot yield was increased (8‒13%) by the soil mulch in two of the three comparisons (Table 6 and Appendix A). Thus, polyethylene mulch was effective also in Site 2, where experiments were carried out in walk-in tunnel greenhouses.

### 2.4. Plant Spacing

The effect of plant density was evaluated at both locations. We compared a density of 24–30 plants/m^2^, which is commonly used in both areas, with a reduced density of 14–15 plants/m^2^. In the greenhouses at Site 1, significant reductions in SBDM were observed in all of the reduced-density plots (Table 7 and Appendix A). The reduction in SBDM severity ranged between 32% and 68%. AUDPC was reduced by 22% to 63% (Table 7 and Appendix A). The relationship between the SBDM severity data and the AUDPC data was calculated. The relation between the severity of disease at each evaluation date and the calculated % disease reduction at the evaluation date or between the values of AUDPC and the calculated % disease reduction were assessed. A negative correlation was found between the severity of disease and % reduction in both cases, single-day disease level, or AUDPC (*p* < 0.01 in both cases; Table 7), pointing to the fact that under lower disease pressure, the less-dense plant spacing provides more pronounced disease suppression. In three experiments, yield was decreased by 12% to 31% by sparse planting compared to dense planting (Table 7 and Appendix A).

The reduced planting density, which was examined in six experiments at Site 2, resulted in a significant (19.5–52.5%) reduction in SBDM severity in six of the eight evaluations. AUDPC was reduced by 21.7% to 50.8%. The relationship between SBDM severity and disease reduction was negative (*p* < 0.1, Table 8). No effect on yield was recorded in the two experiments in which we measured yield (Table 8 and Appendix A). Thus, in both locations, disease severity was reduced by diluted planting density, but yield was not increased.

## 3. Discussion

Cultural measures can be used to suppress SBDM. The following factors were examined in the present work: tunnel orientation, air circulation in a greenhouse, polyethylene mulch, and plant spacing. The level of disease control did not vary with the different growing seasons and intermediate-level disease control was observed in both locations. Studies of cultural means of controlling downy mildews are scarce; therefore, it is only possible to compare the currently studied means with similar methods that have been examined in other patho-systems.

Nighttime air circulation in the greenhouse was increased in an effort to reduce the amount of water on the aerial plant organs and the amount of time that water was present. Humidity-promoted pathogens such as *P. belbahrii* thrive in the film of water that primarily appears at night and depend on it for their reproduction and their ability to infect the plant [9,13]. Increased air circulation can reduce the presence of water at the boundary layer over the plant organs even under humid conditions [20] and can limit humidity-promoted diseases [4]. Indeed, the test carried out over three growing seasons revealed that increased air circulation at night can suppress SBDM in greenhouses. Similarly, Papas [21] concluded that *B. cinerea* in out-of-season tomato (*Solanum lycopersicum*) plants grown in unheated glasshouses in Greece can be limited by adequate indoor air circulation. Similarly, in-bed air circulation has been shown to reduce *B. cinerea* gray mold in lisianthus (*Eustoma grandiflorum*) [22].

In the sweet basil-growing regions in Israel, walk-in tunnels are traditionally oriented east-west. The tunnel orientation affects the microclimate in the tunnel, since it relates to the direction of the sun [6,9,23] and prevailing winds. Surprisingly, the north-south orientation was associated with lower SBDM severity in all of our experiments. Nevertheless, the yields from north-south and east-west tunnels were similar. These results suggest that the temperature × hours gain in the two types of tunnels are similar, but the moisture duration was different. We are not aware of similar results in other patho-systems.

Polyethylene soil cover has been suggested for the control of gray mold induced by *B. cinerea* [24]. A plastic cover with a white upper surface reduced the incidence of *B. cinerea* infection in strawberry (*Fragaria × ananassa*), compared to bare soil [25]. Similarly, in lisianthus, the use of a polyethylene barrier between the lower leaves and soil that prevented the lower leaves from coming into contact with the wet soil reduced the development of *B. cinerea* along the leaves toward the stem and prevented plant mortality [22]. The prevention of contact between the canopy and the wet soil is not relevant in the case of SBDM. However, polyethylene mulch effectively suppressed SBDM in both locations, congruent with the effects such mulch has been shown to have on *B. cinerea*-induced gray mold [8] and *S. sclerotiorum*-induced white mold [7] in sweet basil. As expected [7,8], polyethylene mulch increased shoot yield in two of three experiments.

In addition to decreasing evaporation from the growth medium, the polyethylene mulch also increases the bed temperature. Shtienberg et al. [26] showed that the polyethylene mulch causes irradiation flux across the canopy and the drying of the leaves and fruits of greenhouse tomato and cucumber (*Cucumis sativus*) plants. This helps to control tomato late blight (*Phytophthora infestans*) and downy mildew (*Pseudoperonospora cubensis*) in cucumber [26]. Polyethylene soil cover is associated with increased yields, thanks to the accumulation of heat in the root zone and higher soil temperatures [27]. It was suggested that passive greenhouse warming increases sweet basil’s resistance to downy mildew by warming the root zone [9]. Recently, Gupta et al. [28] demonstrated that warming the root zone induces systemic resistance in plants. Indeed, warming the root zone of sweet basil under field conditions resulted in shoot resistance to the necrotrophic fungi *B. cinerea* and *S. sclerotiorum* that continued after harvest [7,8]. It may be assumed that polyethylene mulch may reduce the susceptibility of harvested leaves also to SBDM.

Reducing the density of sweet basil plants reduced SBDM, as previously demonstrated for gray mold and white mold [7,8]. Reduced plant density has also been shown to suppress disease in other patho-systems. Vieira et al. [29] reported decreased incidence of white mold and increased soybean yields when within-row densities were reduced. Lower plant density has also been shown to reduce stem gray mold in lisianthus [22]. Reducing the number of blond psyllium (*Plantago ovata*) seeds sown per unit area reduced the incidence of downy mildew (*Peronospora alta*) in that crop [30]. The reduction from 12 to 6 bean plants/m^2^ decreased the severity of bean white mold (*S. sclerotiorum*) in one of two experiments, but did not decrease yield [31]. The severity of soybean stem canker (*Diaporthe phaseolorum* var. *meridionalis*) decreased proportionately to a decrease in plant densities [32]. In downy mildew of rose (*Peronospora sparsa*), reducing the density of container-grown plants had a measurable effect on the progress of downy mildew [33].

The reduction in sweet basil planting density resulted in reduced canopy volume at the beginning of the season. But, after the second harvest, the canopy was dense. Nevertheless, SBDM levels were lower in the reduced planting despite the dense canopy. The mechanism of this control could not be studied with the biotroph *P. belbahrii*, but our experience with *B. cinerea* and *S. sclerotiorum* pointed to reduced shoot susceptibility to pathogens [7,8]. We hypothesize that the reduction in planting density also affects the plants’ susceptibility to SBDM.

We calculated the correlation between disease severity values and the intensity of disease reduction across experiments. There was a negative correlation between the disease in the denser plots and disease reduction in the plots in which plants were planted at the lower density, pointing to the fact that under conditions of lower disease pressure, increased plant spacing provides more pronounced disease suppression. Surprisingly, such a negative correlation was not found for the polyethylene mulch practice. In some experiments, we also examined the possibility of combining polyethylene mulch with reduced planting density, but that combination did not provide synergistic disease control (results not shown). When applied in combination with chemical fungicide, neither reduced plant density nor the use of polyethylene provided synergistic disease control (results not shown). As described, there was also no synergistic effect between increased air circulation and transparent or gray polyethylene mulch.

## 4. Materials and Methods

Experiments were carried out at two experimental stations (Sites 1 and 2, described in detail below) under semi-commercial conditions during the years 2013–2015. Sweet basil cv. Peri [34] plants were used in all of these experiments. Plugs were prepared in a commercial nursery (Hishtil, Ashkelon, Israel) and transplanted 3 to 4 weeks after seeding. Each plug contained 3 to 5 plants, but the plugs are usually referred to as plants. “Peri” is susceptible to *P. belbahrii* [9]. The experiments were carried out in greenhouses (Site 1) and in walk-in tunnels (Site 2). Downy mildew epidemics occurred naturally at the field sites, following the placement of infected basil plants next to the plots as described below.

### 4.1. Inoculation with P. belbahrii and Disease Evaluation

Spores of *P. belbahrii* were harvested in water by washing sporulating leaves of sweet basil plants that were kept in an experimental greenhouse at the Volcani Center, Agricultural Research Organization, Israel. The suspension was then filtered through cheesecloth. The concentration of spores was determined using a hemocytometer and a light microscope, and adjusted to 1 × 10^3^ cells ml^−1^. Potted sweet basil plants were inoculated by spraying with a spore suspension (5 mL plant^−1^), incubated at high RH (>95%) in the dark in a growth chamber at 22 ± 1 °C for 12 h and then incubated in a greenhouse chamber at 22 ± 2 °C for 1 week, and incubated at high RH (>95%) in the dark in a growth chamber at 22 ± 1 °C for 12 h and then incubated in a greenhouse chamber at 22 ± 2 °C for symptom development [9]. The potted sweet basil plants subjected to this artificial inoculation served as a source of inoculum to ensure even inoculum loads across the greenhouses and walk-in tunnels. The plants were placed at the borders of each plot.

The evaluation of the severity of sweet basil downy mildew (SBDM) in the plots included all plants except those along the 1 m edges of each plot. The severity of SBDM was determined periodically in all plants of each plot in each experiment on a scale of 0 to 100, in which 0 = all plants visually healthy, 10 = 10% of the leaf area in the plot covered by typical downy mildew symptoms of chlorosis and/or dry necrotic lesions or *P*. *belbahrii* spores on the undersides of the leaves, and 100 = all leaves on all plants in the plot show typical downy mildew symptoms/signs [9].

### 4.2. Shoot Weight

In selected experiments, shoots longer than 15 cm were harvested and weighed three to five times during the growing season, as detailed below. The yield was collected separately for each plot, sorted for quality, and calculated per m^2^ bed. The cumulative yield figures for the various harvests were calculated and those figures are presented.

### 4.3. Site 1—Eden Experimental Station

Experiments were conducted at the Eden Experimental Station (32°46′79 N, 35°48′88 E; 120 m below mean sea level) at the Emek Hamaayanot Research and Development Center. The regional climate is Mediterranean, semiarid with winter rains and a dry, hot summer. At this site, experiments were carried out in two 400 m^2^ greenhouses. The structures were covered with 150 µm-thick Sunsaver Clear IR AV polyethylene (Ginegar Plastic Products, Kibutz Ginegar, Israel). The greenhouses were aerated during the day and closed during the night (18:00 to 07:00). At night, the greenhouses were heated to 12 °C to prevent physiological damage to the leaves. There were five bays in each greenhouse and the bays were separated with 1.8 m-high transparent polyethylene.

The potting material was tuff (volcanic gravel; 3 to 6 mm particles) placed in plastic containers that were 1 m wide × 15 cm deep × 20 m long (Mapal, Mevo Hama, Israel). Plants were irrigated daily according to local extension service recommendations, allowing 30% drainage, and fertigated proportionally with 5-3-8 N-P-K fertilizer at a rate of 2 L/1000 L water. The nutrient concentrations were therefore 8.6, 1.0, and 4.0 mM N, P, and K, respectively. Fertigation was performed using a 17 mm drip-irrigation pipe with a 1 L/h dripper embedded in the pipe every 20 cm. Plots were 5 m long each, containing 108–125 plants/plot at the higher plant density (24–25 plants/m^2^) mentioned below.

Experiments were carried out over three consecutive growing seasons, with planting dates of 9 September (fall 2013), 24 February (spring 2014), and 19 January (winter 2015). Treatments consisted of different cultural methods, as detailed below and in Table 9. Plots consisted of one bed (1 m wide and 4.5 to 5.0 m long) and there were 4–8 plot replicates.

Fall 2013 experiments: SBDM was first observed on 27 November 2013, 80 days after planting. There were five shoot harvests, starting 7 October 2013 (29 days after planting).

Spring 2014 experiments: SBDM was first observed on 20 March 2014, 24 days after planting. There were six shoot harvests, starting 26 March 2014 (30 days after planting).

Winter 2015 experiments: SBDM was first observed on 14 April 2015, 90 days after planting. Shoots were harvested five times, starting 1 March (45 days after planting).

### 4.4. Cultural Methods Applied at Site 1

Air circulation: Four fans (60 cm diam., Adirom Heating and Ventilation Engineering Ltd., Ashkelon, Israel) were installed 2 m above the beds, facing the canopy of plants that were planted at the area of one third of a greenhouse bay toward the north or south edge of the greenhouse. The fans were operated once every hour for 15 min from 19:00 until 08:00. Beds were covered with gray polyethylene mulch and the planting density was 24–25 plants/m^2^.

Planting density: Sweet basil plants were planted at two densities: 24–30 plants/m^2^, as is customary in the area, and 14–15 plants/m^2^. The higher planting density was also used in experiments in which planting density was not a tested parameter. The beds were left bare or covered with gray or transparent polyethylene (Table 9).

Polyethylene mulch: The beds were either left uncovered (bare growing medium) or covered with sheets of polyethylene. Several types of polyethylene were examined: (1) transparent 30 µm-thick Sunsaver Clear IR polyethylene (Ginegar), (2) gray-black 30 µm-thick Mulch-More polyethylene (Ginegar) with the gray-colored side visible and the black-colored side facing the ground, and (3) yellow-brown 30 µm-thick Mulch-More polyethylene (Ginegar) with the yellow side visible and the brown side facing the ground. The plant density was 24–25 plants/m^2^ (Table 9).

### 4.5. Site 2—Zohar Experimental Station

This research station is located in the Sedom area south of the Dead Sea and is part of the Northern Arava Research and Development Center. It is located at 30°94′656.2 N, 35°40′341.7 E at 354 m below mean sea level. The weather at the Zohar Station is arid. In the winter, rain is rare and the mean daytime temperature is 22 °C. The summers are dry and hot, with an average daily temperature of 33 °C. The work at the Zohar Experimental Station was carried out in 10 walk-in tunnels. Each tunnel was 40 m long and 5 m wide (200 m^2^). The structures were covered with 100 µm-thick Sunsaver Clear IR AV polyethylene (Ginegar Plastic Products, Ginegar, Israel). The front and back openings of each tunnel were covered with 50-mesh netting. Five round aeration openings (50 cm diam.) were cut along the length of the tunnels and covered with 50-mesh netting. One-meter-wide sandy soil beds were planted with 30 plants/m^2^, unless otherwise noted. Plants were irrigated with local brackish water (4 decisiemens per meter), according to the local extension service recommendations, and fertigated with 1.0 L/1000 L 8-2-4 N-P-K fertilizer. Nutrient concentrations were therefore 6.9, 0.33, and 1.0 mM N, P, and K, respectively. Fertigation was performed using a 17 mm drip-irrigation pipe with a 1.2 L/h dripper embedded in the pipe every 20 cm. Each plot consisted of two beds that were each 9 m long, unless otherwise mentioned.

Experiments were carried out over two consecutive growing seasons, with planting dates of 19 February (spring 2014 season) and 11 November 2014 (winter 2015 season). Treatments consisted of different cultural methods, as detailed below and in Table 10. Plots consisted of two beds (1 m wide and 9 m long) containing 270 plants/plot of 30 plants/m^2^ and there were 5‒10 plot replicates.

Spring 2014 experiments: SBDM was first observed on 20 March 2014, 31 days after planting. There were six shoot harvests, beginning 15 March 2014 (36 days after planting).

Winter 2015 experiments: SBDM was first observed on 21 January 2015, 71 days after planting. Shoots were harvested four times starting 8 December 2014 (27 days after planting).

### 4.6. Cultural Methods Applied at Site 2

Tunnel direction: Walk-in tunnels were oriented north-south or east-west with either bare soil or with transparent polyethylene mulch. The planting density in the plots was 30 plants/m^2^ (Table 10).

Planting density: Sweet basil plants were planted at two densities: 30 plants/m^2^, as is the common local practice, or 15 plants/m^2^. The higher planting density was also used in experiments in which the planting density was not a tested parameter. The soil was covered with transparent polyethylene (Table 10).

Polyethylene soil mulch: The beds were either left uncovered (bare growing medium) or covered with transparent polyethylene (30 µm-thick Sunsaver Clear IR polyethylene; Ginegar). The beds were planted with 30 plants/m^2^ in tunnels oriented either north-south or east-west (Table 10).

### 4.7. Experimental Design and Statistical Analysis

Treatments in each year and each field experiment were replicated 4–10 times. Replicates of each treatment were arranged randomly. Disease severity was evaluated in each plot (replicate). Area under the disease severity progress curve (AUDPC) values were also calculated. Data in percentages were arcsine-transformed before further analysis. Disease severity (%) and AUDPC (% × days) data were analyzed using ANOVA and Tukey’s HSD test. Standard errors (SE) of the means were calculated and disease levels were statistically separated following a one-way analysis of variance. Treatments in experiments with combined two-treatment factors were statistically separated following a two-way analysis of variance. Statistical analyses were performed using JMP 5.0 software (SAS Institute, Cary, NC, USA).

Disease reduction was calculated as follows:% disease reduction = 100 − 100 × (disease severity_TT_/disease severity_control_).(1)

The combined effect of the control measures used was estimated using the Abbott formula [35,36]. The expected disease reduction (control efficacy) and the combined suppressive activity were calculated as:CE_exp_= a + b − a × b/100 and SF = CE_obs_/CE_exp_,(2)
where a = disease reduction due to one measure when applied alone, b = disease reduction due to the other measure when applied alone, CE_exp_ = expected control efficacy of the combined treatment if the two measures act additively, CE_obs_ = observed disease reduction for the combined treatment, and SF = the synergy factor achieved by the combined treatment. When SF = 1, the interaction between the control measures is additive. When SF < 1, the interaction is antagonistic, and when SF > 1, the interaction is synergistic [26,35,36]. The same formula was used to calculate SF in the context of yield.

## 5. Conclusions

Increased air circulation, reduced plant density, polyethylene mulch, and the north-south orientation of walk-in tunnels moderately reduced SBDM under commercial conditions. These practices can contribute to efforts to reduce the dependence on chemical fungicides in sweet basil crops that commercially, because of demands of minimized chemical residues, can tolerate only limited use of such chemicals at application times that are temporally far from harvest.

## Figures and Tables

**Figure 1 plants-10-00907-f001:**
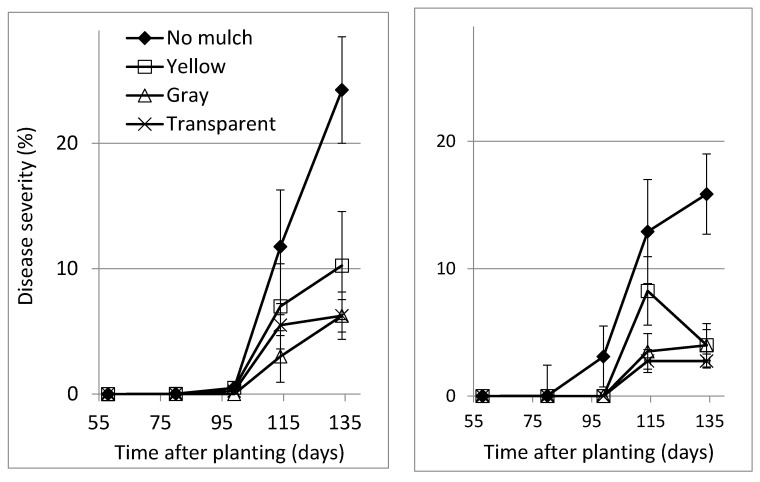
Effect of the type of polyethylene mulch on the development of sweet basil downy mildew (SBDM) on plants grown in tuff growing medium at Site 1 (experiment M8). Plants were grown with no supplemental air circulation (**left**) or with extra air circulation (**right**). SBDM severity was evaluated on a 0–100% scale, in which 0 = healthy plants and 100% = plants completely covered by SBDM symptoms/signs. SBDM severity is presented as percentage ± SE. An analysis of these results is presented in Table 5.

**Table 1 plants-10-00907-t001:** Effect of tunnel orientation on the severity of sweet basil downy mildew (SBDM) in walk-in tunnels at Site 2.

Experiment	AUDPC over the Growing Season (% × Days) ^b^			Cumulative Yield (g/m^2^)		
	Period (Days)	East-West	North-South	Period (Days)	East-West	North-South
TD1	114	2447 ± 232.3 ^a^	1834 ± 213.6 ^b^	93	3280 ± 327.4 ^a^	3213 ± 422.1 ^a^
TD2	114	3063 ± 272.3 ^a^	2053 ± 260.9 ^b^	93	2973 ± 164.4 ^a^	3062 ± 154.9 ^a^
TD3	112	3605 ± 68.2 ^a^	1327 ± 122.9 ^b^	80	1843 ± 210.4 ^a^	2107 ± 192.5 ^a^

^a^ Experiments were carried out with sweet basil plants grown in sand in walk-in tunnels during spring 2014 (experiments TD1 and TD2) and winter 2014–2015 (experiment TD3) growing seasons. Beds were covered with transparent polyethylene (TD1 and TD3) or left uncovered (TD2). ^b^ SBDM severity was evaluated on a 0–100% scale, in which 0 = healthy plants and 100% = plants completely covered by SBDM symptoms/signs, and the value of the area under disease progress curve was calculated for each replicate. Averages ± SE. Values in each pair followed by a different letter are significantly different according to one-way ANOVA with Tukey’s HSD. Default significance levels were set at α = 0.05.

**Table 2 plants-10-00907-t002:** Effect of increased air circulation on the severity of sweet basil downy mildew (SBDM) in greenhouses at Site 1a.

Experiment	AUDPC over the Growing Season (% × Days) ^b^			Cumulative Yield (g/m^2^)		
	Period (Days)	Without Fans	With Fans	Period (Days)	Without Fans	With Fans
AC1	134	838.9 ± 75.42 ^a^	311.8 ± 85.12 ^b^			
AC2	109	223.0 ± 42.4 ^a^	61.3 ± 20.8 ^b^	91	3381 ± 179.5 ^a^	3327 ± 214.1 ^a^
AC3	151	1151 ± 93.6 ^a^	730 ± 97.1 ^b^	91	2820 ± 279.5 ^a^	2420 ± 234.1 ^a^

^a^ Field experiments were carried out with sweet basil plants grown in detached growth medium covered by gray polyethylene at Site 1 during the autumn 2013, spring 2014, and winter 2015 seasons (experiments AC1, AC2, and AC3, respectively). ^b^ SBDM severity was evaluated on a 0–100% scale, in which 0 = healthy plants and 100% = plants completely covered by SBDM symptoms/signs, and the value of the area under disease progress curve was calculated for each replicate. Averages ± SE. Values in each pair followed by a different letter are significantly different according to one-way ANOVA with Tukey’s HSD. Default significance levels were set at α = 0.05.

**Table 3 plants-10-00907-t003:** Effect of polyethylene mulch on the severity of sweet basil downy mildew (SBDM) and yield of sweet basil in greenhouses at Site 1a.

Experiment	AUDPC over the Growing Season (% × Days) ^b^			Cumulative Yield (g/m^2^)		
	Period (Days)	No Mulch	Polyethylene	Period (Days)	No Mulch	Polyethylene
M2	114	554.6 ± 67.03 ^a,c^	272.2 ± 68.95 ^b^	100	4570 ± 108.8 ^b^	4862 ± 99.3 ^a^
M4	93	1005 ± 118.3 ^a^	866.7 ± 97.04 ^b^	94	3042 ± 108.7 ^b^	3628 ± 125.8 ^a^
M5	93	288.0 ± 60.44 ^a^	197.0 ± 33.42 ^a^	83	2776 ± 154.8 ^a^	3523 ± 170.8 ^a^
M6	123	2820 ± 211.7 ^a^	1012 ± 143.7 ^b^	91	2820 ± 211.7 ^b^	4013 ± 143.7 ^a^
M7	123	1089 ± 98.8 ^a^	572.8 ± 82.33 ^b^			
	151	1802 ± 120.3 ^a^	1031 ± 125.6 ^b^	91	1010 ± 420.3 ^b^	2820 ± 402.5 ^a^

^a^ Field experiments were carried out with sweet basil plants grown in detached growth medium at Site 1 during the autumn 2013 (M2), spring 2014 (M4–M6), and winter 2015 (M7) growing seasons. Beds were covered with gray polyethylene (M2, M4, and M5) or transparent polyethylene (M6 and M7). ^b^ SBDM severity was evaluated on a 0‒100% scale, in which 0 = healthy plants and 100% = plants completely covered by SBDM symptoms/signs, and the value of the area under disease progress curve was calculated for each replicate. ^c^ Averages ± SE. Values in each pair followed by a different letter are significantly different according to one-way ANOVA with Tukey’s HSD. Default significance levels were set at α = 0.05.

**Table 4 plants-10-00907-t004:** Effect of the type of polyethylene mulch on the severity of sweet basil downy mildew (SBDM) in greenhouses at Site 1a.

Polyethylene Mulch	Experiment M1			Experiment M3	
	Severity (%) ^b^ at Day 114	Severity (%) at Day 131	AUDPC (% × Days) Over 131 days	Severity (%) at Day 93	AUDPC (% × Days) Over 93 Days
None	13.4 ± 2.09 ^a,c^	21.5 ± 2.66 ^a^	378.8 ± 40.64 ^a^	13.1 ± 2.29 ^a^	318.3 ± 53.37 ^a^
Yellow	7.6 ± 2.02 ^a,b^	7.1 ± 2.38 ^b^	205.1 ± 54.64 ^b^	4.6 ± 1.36 ^b^	228.1 ± 34.99 ^a,b^
Gray	3.8 ± 1.16 ^b^	4.0 ± 1.37 ^b^	103.4 ± 27.96 ^b^	7.5 ± 1.13 ^b^	167.8 ± 66.31 ^b^
Transparent	4.1 ± 0.83 ^b^	4.5 ± 100 ^b^	117.2 ± 22.43 ^b^	6.5 ± 2.73 ^b^	149.6 ± 21.70 ^b^

^a^ Experiments were carried out with sweet basil plants grown in detached growth medium at Site 1 during the autumn 2013 (experiment M1) and spring 2014 (experiment M3) growing seasons. ^b^ SBDM severity was evaluated on a 0–100% scale, in which 0 = healthy plants and 100% = plants completely covered by SBDM symptoms. ^c^ Averages ± SE. Values in each pair followed by a different letter are significantly different according to one-way ANOVA with Tukey’s HSD. Default significance levels were set at α = 0.05.

**Table 5 plants-10-00907-t005:** Results from experiment M8 (Figure 1).

**Mulch (M) Treatment**	**Severity Parameters and Air Circulation (AC)**			
	**Day 134 (% Severity)**		**AUDPC, 134 Days (% × Days) ^a^**	
	**Without AC**	**With AC**	**Without AC**	**With AC**
No mulch	24.3 ± 4.42	15.9 ± 3.41	458.7 ± 79.81	437.5 ± 79.81
Yellow	10.3 ± 4.11	4.0 ± 2.11	234.2 ± 53.24	185.3 ± 53.24
Gray	6.2 ± 1.23	4.0 ± 1.89	115.0 ± 14.06	101.3 ± 14.06
Transparent	6.3 ± 1.32	2.8 ± 0.41	159.3 ± 15.96	75.6 ± 15.96
AC × M interaction	No		No	
**Major-treatments analysis ^b^**				
**M treatments**	**Day 134 (% severity)**	**AUDPC, 134 days (% × days)**		
No mulch	20.1 ^a,c^	448.1 ^a^		
Yellow	7.2 ^b^	209.7 ^b^		
Gray	5.1 ^b^	108.1 ^c^		
Transparent	4.5 ^b^	117.4 ^c^		
**AC treatments**				
Without	11.8 ^a^	241.8 ^a^		
With	6.7 ^b^	199.9 ^b^		

^a^ The area under disease progress curve (AUDPC) over 134 days was calculated. ^b^ The interaction between [M treatments] × [AC treatments] did not significantly affect severity or AUDPC, so each major parameter was analyzed independently of the other. ^c^ M and AC treatments in each column followed by a different letter are significantly different according to two-way ANOVA with Tukey’s HSD. Default significance levels were set at α = 0.05.

**Table 6 plants-10-00907-t006:** Effects of polyethylene mulch on the severity of sweet basil downy mildew (SBDM) and yield of sweet basil in walk-in tunnels at Site 2a.

Experiment	AUDPC over the Growing Season (% × Days) ^b^			Cumulative Yield (g/m^2^)		
	Period (Days)	No Mulch	Polyethylene	Period (Days)	No Mulch	Polyethylene
M8	114	2131 ± 128.5 ^a,c^	1047.8 ± 111.5 ^b^	110	3425 ± 128.8 ^a^	3473 ± 95.5 ^a^
M9	114	3063 ± 242.4 ^a^	1833.5 ± 197.6 ^b^	110	3359 ± 122.7 ^b^	3802 ± 146.4 ^a^
M10	114	2054 ± 124.7 ^a^	1615.4 ± 112.4 ^b^	101	4628 ± 139.7 ^b^	4986 ± 117.2 a
M11	112	2040 ± 243.1 ^a^	1708 ± 147.5 ^a^			
M12	112	2863 ± 212.7 ^a^	2369 ± 224.1 ^b^			
M13	112	3156 ± 204.3 ^a^	2319 ± 243.7 ^b^			
M14	112	197.6 ± 47.98 ^a^	102.6 ± 27.64 ^b^			

^a^ Experiments were carried out with sweet basil plants grown in detached growth medium at Site 2 during the spring 2014 (M8–M10) and winter 2014–2015 (M11–M14) growing seasons. Mulch-treatment plots were covered with transparent polyethylene. ^b^ SBDM severity was evaluated using a 0–100% scale, in which 0 = healthy plants and 100% = plants completely covered by SBDM symptoms/signs, and the value of the area under disease progress curve was calculated for each replicate. ^c^ Averages ± SE. Values in each pair followed by a different letter are significantly different according to one-way ANOVA with Tukey’s HSD. Default significance levels were set at α = 0.05.

**Table 7 plants-10-00907-t007:** Effect of planting density (PD) on the severity of sweet basil downy mildew (SBDM) in greenhouses at Site 1a.

Experiment	AUDPC over the Growing Season (% × Days) ^b^			Cumulative Yield (g/m^2^)		
	Period (Days)	Dense (24)	Sparse (14)	Period (Days)	Dense (24)	Spars (14)
PD1	114	654.6 ± 62.42 ^a,c^	383.3 ± 28.75 ^b^			
PD2	114	48.0 ± 3.53 ^a^	17.5 ± 3.72 ^b^	100	4570 ± 158.9 ^a^	3985 ± 162.4 ^b^
PD3	134	169.1 ± 20.14 ^a^	63.9 ± 18.05 ^b^			
PD4	93	1053 ± 48.6 ^a^	821 ± 70.5 ^b^	91	3110 ± 167.8 ^a^	2130 ± 224.9 ^b^
PD5	93	1005 ± 118.3 ^a^	664 ± 79.8 ^b^			
PD6	151	805 ± 74.6 ^a^	503 ± 69.3 ^b^	128	2426 ± 109.6 ^a^	2140 ± 107.1 ^b^
PD7	151	909 ± 81.3 ^a^	604 ± 71.1 ^b^			
**Disease severity—Disease reduction relation**						
		Equation	y = −25.4x + 1712			
		*r*	0.9766			
		*n*	5			
		*p*	<0.01			

^a^ Experiments were carried out with sweet basil plants grown in detached growth medium in greenhouses at Site 1 during the autumn 2013 (experiments PD1‒PD3), spring 2014 (experiments PD4–PD6), and winter 2015 (experiment PD7) growing seasons. Beds were left bare (experiments PD1, PD4, and PD6) or covered with polyethylene (experiments PD2, PD3, PD5, and PD7). ^b^ SBDM severity was evaluated using a 0–100% scale, in which 0 = healthy plants and 100% = plants completely covered by SBDM symptoms, and the value of the area under disease progress curve was calculated for each replicate. ^c^ Averages ± SE. Values in each pair followed by a different letter are significantly different according to one-way ANOVA with Tukey’s HSD. Default significance levels were set at α = 0.05. Equations for the relation between disease severity values of dense PD and significant disease reduction values by the sparse PD are presented and the Pearson regression value (*r*) is presented along with the significance levels (*p*).

**Table 8 plants-10-00907-t008:** Effect of planting density (PD) on the severity of sweet basil downy mildew (SBDM) in walk-in tunnels at Site 2a.

Experiment	AUDPC over the Growing Season (% × Days) ^b^			Cumulative Yield (g/m^2^)		
	Period (Days)	Dense (30)	Sparse (15)	Period (Days)	Dense (30)	Sparse (15)
PD8	98	2131 ± 128.5 ^a^	1048 ± 111.5 ^b^	110	3581 ± 116.4 ^a^	3317 ± 98.7 ^a^
PD9	114	2776 ± 331.9 ^a^	2121 ± 252.4 ^b^	110	3652 ± 242.4 ^a^	3508 ± 221.3 ^a^
PD10	114	2058 ± 164.7 ^a^	1611 ± 147.7 ^b^			
PD11	98	197.6 ± 51.42 ^a^	108.5 ± 22.44 ^b^			
PD12	112	3156 ± 232.1 ^a^	2319 ± 117.4 ^b^			
PD13	112	3254 ± 212.4 ^a^	2421 ± 112.1 ^b^			
**Disease severity—AUDPC relation**						
			insignificant			

^a^ Experiments were carried out with sweet basil plants grown in sand in walk-in tunnels at Site 2 during the spring 2014 (experiments PD8‒PD10) and winter 2014–2015 (experiments PD11–PD13) growing seasons. Beds were covered with polyethylene. ^b^ SBDM severity was evaluated using a 0–100% scale, in which 0 = healthy plants and 100% = plants completely covered by SBDM symptoms/signs, and the value of the area under disease progress curve was calculated for each replicate. Averages ± SE. Values in each pair followed by a different letter are significantly different according to one-way ANOVA with Tukey’s HSD. Default significance levels were set at α = 0.05.

**Table 9 plants-10-00907-t009:** Experiments conducted with sweet basil downy mildew in greenhouses at Site 1.

Growing Season	Expt. No.	Tested Factors	Other Agronomic Parameters		
			Polyethylene Mulch	Plant Spacing (Plugs/m^2^)	Air Circulation (AC)
Fall 2013	M1	Polyethylene mulch (PM)	Gray, yellow, transparent PE	25	No
	M2	PM	Gray PE	25	No
	PD1	Planting density (PD)	Bare soil	25 vs. 15	No
	PD2	PD	Gray PE	30 vs. 15	No
	PD3	PD	Gray PE	25 vs. 15	No
	AC1	AC	Gray PE	25	AC
Spring 2014	M3	PM	Gray, yellow, transparent PEs	24	No
	M4, M5	PM	Gray PE	24	No
	M6	PM	Transparent PE	24	No
	PD4, PD6	PD	Bare soil	21 and 24 vs. 14	No
	PD5	PD	Gray PE	24 vs. 14	No
	AC2	AC	Gray PE	24	AC
Winter 2015	M7	PM	Transparent PE	24	No
	M8	PM	Gray, yellow, transparent PEs	24	AC
	PD7	PD	Gray PE	24 vs. 14	No
	AC3	AC	Gray PE	24	AC

**Table 10 plants-10-00907-t010:** Experiments conducted with sweet basil downy mildew in walk-in tunnels at Site 2.

Growth Season	Expt. No.	Tested Factors	Other Agronomic Parameters		
			Polyethylene Mulch	Plant Spacing (Plugs/m^2^)	Tunnel Direction (TD)
Spring 2014	M8, M10	Polyethylene mulch (PM)	Transparent vs. bare	30	East-West (EW)
	M9	PM	Transparent vs. bare	30	North-South (NS)
	PD8, PD9	Plant density (PD)	Transparent	30 vs. 15	EW
	PD10	PD	Transparent	30 vs. 15	NS
	TD1	Tunnel direction (TD)	Transparent	30	NS vs. EW
	TD2	Tunnel direction (TD)	Bare	30	NS vs. EW
Winter 2014‒2015	M11, M14	PM	Transparent vs. bare	30	NS
	M12, M13	PM	Transparent vs. bare	30	EW
	PD11	PD	Transparent	15 vs. 30	NS
	PD12, PD13	PD	Transparent	15 vs. 30	EW
	TD3	TD	Transparent	30	NS vs. EW

## Data Availability

The data that support the findings of this study are available from the corresponding author upon reasonable request.

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
