# Peer review of "Effects of Agronomic Practices on the Severity of Sweet Basil Downy Mildew (Peronospora belbahrii)"

_plants, 2021, doi:10.3390/plants10050907_

Round 1

Reviewer 1 Report

The authors of the manuscript entitled “Effects of Agronomic Practices on the Severity of Sweet Basil Downy Mildew (Peronospora belbahrii)” reported the effect of several cultural methods on disease severity caused by Peronospora belbahrii on sweet basil. The authors also measured the impact on yield of part of these cultural methods.

I enjoyed reading the manuscript because its fluency of writing. The topic is interesting because the economic importance of sweet basil downy mildew in Israel and in the Mediterranean basin. The Manuscript provides an advancement in the field of research, with a focus in the sustainable management of P. belbahrii under commercial conditions in greenhouses and walk-in tunnels. Study design and methodology are appropriate although some minor concerns need to be addressed:

  • Did you perform statistical analysis of cumulative data belonging to the same treatment across different years?

Another concern is represented by table formatting. Tables are filled with too much information making them difficult to read. To increase the clarity of the tables I would remove reduction, period and time after planting (this information could be inserted in the text). It would be more useful to report the cumulative data in the table instead of the same experiment in different years. In some cases the readability could be improved by inverting the treatments with the experiments as done in Table 4.

Minor concerns:

Line 19: I suggest including “mulch” after “polyethylene”

Line 86: Write AUDPC in full for the first time

Table 1: reduce table width

Table 3: M5 and M7 disease symptoms seem measured at different time points. Isn't it? Can you clarify?

Table 5: In the bottom part of the table the header of the two columns is missing

Table 6 footnotes: When was it carried out M10?

Lines 264-265: “Infection of cut shoots by P. belbahrii could not be performed” could be removed

Materials and methods: How many plants per plot? How many plants used in infection experiments

Author Response

Reviewer 1

The authors of the manuscript entitled “Effects of Agronomic Practices on the Severity of Sweet Basil Downy Mildew (Peronospora belbahrii)” reported the effect of several cultural methods on disease severity caused by Peronospora belbahrii on sweet basil. The authors also measured the impact on yield of part of these cultural methods.

I enjoyed reading the manuscript because its fluency of writing. The topic is interesting because the economic importance of sweet basil downy mildew in Israel and in the Mediterranean basin. The Manuscript provides an advancement in the field of research, with a focus in the sustainable management of P. belbahrii under commercial conditions in greenhouses and walk-in tunnels. Study design and methodology are appropriate although some minor concerns need to be addressed:

  • Did you perform statistical analysis of cumulative data belonging to the same treatment across different years?

YE: Experiments were carried out in different locations, years (growing seasons) greenhouse types and cultivating practices so generalizing by cumulative statistical analysis may be misleading. However, in the text I generalize the effect of each agro-technical practice by testing the range of disease reduction. In most cases, there was a wide range efficacy that by itself point to the irrelevance of cumulative analysis. At the end of each results paragraph I added a general conclusion that can summarize the effect of each technic that was tested.

Another concern is represented by table formatting. Tables are filled with too much information making them difficult to read. To increase the clarity of the tables I would remove reduction, period and time after planting (this information could be inserted in the text).

YE: The reduction was removed from the tables in their new format. Since the disease severity columns were recommended below to be moved to the supplement, I moved to the table supplements also all columns with the calculation of disease reduction and yield increase/decrease. The tables left in the results section as presented now are much reduced in size, more compact and less packed with information. Nevertheless, when presenting AUDPC, the duration of the recorded epidemic is important and is part of the presented disease severity data so I wish to keep it in the tables.

It would be more useful to report the cumulative data in the table instead of the same experiment in different years.

YE: The cumulative results outcomes are brought in the text that is placed before each table. It is important to detail the experiments that were carried out in each site since the results were somewhat variable and were affected by site, growing practice, season etc. so the range of disease severity and of results and of disease reduction is wide and need to be presented in the tables rather than to be shown as cumulative in the tables.

In some cases the readability could be improved by inverting the treatments with the experiments as done in Table 4.

YE: I agree that table 4 is less complicated than the rest of the tables as originally submitted. Tables were made less complicated once we changed it according to the comment below that suggested to put the disease severity data in the supplement.

Minor concerns:

Line 19: I suggest including “mulch” after “polyethylene”

YE: I included 'mulch' as suggested.

Line 86: Write AUDPC in full for the first time

YE: Right. It is written now: area under disease progress curve (AUDPC)

Table 1: reduce table width

YE: Somehow, one column in table 1 got very wide. It is now arranged.

Table 3: M5 and M7 disease symptoms seem measured at different time points. Isn't it? Can you clarify?

YE: Severity at two dates is presented and the times of evaluation is mentioned in the 2nd column (Time after planting (days)).

Table 5: In the bottom part of the table the header of the two columns is missing

YE: Thank you – I placed the headers of the two columns in the bottom part of the table.

Table 6 footnotes: When was it carried out M10?

YE: Clarified in the footnote so M10 also in spring 2014.

Lines 264-265: “Infection of cut shoots by P. belbahrii could not be performed” could be removed

YE: Removed.

Materials and methods: How many plants per plot? How many plants used in infection experiments

YE: The plants numbers are mentioned now in the M&M general details sections of site 1 and site 2 as plants/plot

Reviewer 2 Report

Dear Authors,

The research work presented in the manuscript is original and with significant importance. However, for acceptance major changes may be need in the results as stated below.

The results should be clearly and concisely. The manuscript has many tables. As many experiments have been carried out, I suggest highlighting important findings, presenting only the most relevant results in the text. The other results I suggest include in the supplementary material. For example, authors should present only AACPC data, since AUDPC is a summary of disease intensity over time. As the author does not discuss the progress of the disease over time, the severity data becomes unnecessary and may be include in the supplementary material.

Author Response

Reviewer 2

Dear Authors,

The research work presented in the manuscript is original and with significant importance. However, for acceptance major changes may be need in the results as stated below.

The results should be clearly and concisely.

YE: Changes were made as answered below. The results are now summarized in each agrotechnical subject so it was made more clear.

The manuscript has many tables. As many experiments have been carried out, I suggest highlighting important findings, presenting only the most relevant results in the text. The other results I suggest include in the supplementary material. For example, authors should present only AACPC data, since AUDPC is a summary of disease intensity over time. As the author does not discuss the progress of the disease over time, the severity data becomes unnecessary and may be include in the supplementary material.

YE: Disease severity was moved to supplementary tables. AUDPC was left in the tables of the main text. Two missing AUDPC data were added in table 7 and one was added in table 8.

Reviewer 3 Report

In my opinion, this manuscript is interesting and well written both in terms of content and language. There is always something to be done better, however, this work is thoughtful and well written. For this reason, I have no comments or queries regarding this manuscript. Congratulations to the authors.

Author Response

Reviewer 3

In my opinion, this manuscript is interesting and well written both in terms of content and language. There is always something to be done better, however, this work is thoughtful and well written. For this reason, I have no comments or queries regarding this manuscript. Congratulations to the authors.

YE: No corrections asked - Thank you.

Round 2

Reviewer 2 Report

Dear authors,

The manuscript plants-1191251 entitled "Effects of agronomic practices on the severity of sweet basil downy mildew (Peronospora belbahrii)" are appropriate for publication.